# Inflammatory Markers and Thromboembolic Risk in Patients with Non-Muscle-Invasive Bladder Cancer

**DOI:** 10.3390/jcm10225270

**Published:** 2021-11-12

**Authors:** Daniel Balan, Mihai Dorin Vartolomei, Annamária Magdás, Noemi Balan-Bernstein, Septimiu Toader Voidăzan, Orsolya Mártha

**Affiliations:** 1Department of Urology, University of Medicine, Pharmacy, Science and Technology “G.E.Palade” of Targu-Mures, 540142 Targu-Mures, Romania; balan_dani@yahoo.com (D.B.); orsim@hotmail.com (O.M.); 2Urology Department, Vienna General Hospital, 1090 Vienna, Austria; mdvartolomei@yahoo.com; 3Department of Internal Medicine, University of Medicine, Pharmacy, Science and Technology “G.E.Palade” of Targu-Mures, 540142 Targu-Mures, Romania; 4Clinic of Pneumology, Mures County Hospital, 540103 Targu-Mures, Romania; b.noemi47@gmail.com; 5Department of Epidemiology, “G. E. Palade” University of Medicine, Pharmacy, Science, and Technology of Targu Mures, 540142 Targu-Mures, Romania; septimiu.voidazan@umfst.ro

**Keywords:** IMPROVE, neutrophil to lymphocyte ratio, complete blood count, lymphocyte to monocyte ratio

## Abstract

Introduction: Patients with bladder cancer have a high risk of venous thrombosis that represents a key challenge for physicians in the decision-making for initiating anticoagulation therapy. Non-muscle-invasive bladder cancer (NMIBC) represents more than 70% of all diagnosed bladder malignancies; therefore, we aimed to evaluate the relationship of the neutrophil to lymphocyte ratio (NLR), lymphocyte to monocyte ratio (LMR), and risk of thrombosis by using the International Medical Prevention Registry on Venous Thromboembolism (IMPROVE) score as well as the risk of bleeding by using the IMPROVE Bleeding Risk Assessment Score in a study cohort. Material and Methods: This was a retrospective observational study involving 130 patients who met the inclusion criteria: age > 18 years, stage pTa-pT1 NMIBC. The exclusion criteria were age < 18 years; stage pT2 or higher; or a presentation of metastasis, inflammatory, liver or autoimmune diseases, or other systemic neoplasms. In order to evaluate the risk of thromboembolic events as well as those of bleeding, the IMPROVE scores were calculated for each patient. Subjects were categorized in a Low IMPROVE group (< 4 points) or a High IMPROVE group. By using uni- and multivariate regression models, we analyzed CBC-derived parameters which could be associated with a higher risk of venous thrombosis in subjects with low or high IMPROVE scores. Results: Patients with IMPROVE score greater than 4 were associated with higher NLR, LMR and lymphocyte values (*p* < 0.05). In a multivariate regression model, the IMPROVE score was significantly influenced by lymphocyte count (*p* = 0.007) as well as the NLR value (*p* < 0.0001). Conclusions: In our study population, subjects with NMIBC with low lymphocytes and NLR > 3 were at a higher risk of developing venous thromboembolic events, reflected by an IMPROVE score of greater than 4. The IMPROVE and IMPROVE Bleeding Risk Assessment Scores are easy to use, and, complemented with the CBC-derived lymphocyte to monocyte ratio as a prothrombotic marker, could aid in the decision of prophylactic anticoagulation therapy during admission.

## 1. Introduction

It is well known that patients with malignancies have a high risk of venous thrombosis. However, the risks of thrombosis and those of bleeding often coexist, and represent the main challenge for physicians in decision making for the use of anticoagulation therapy. Complete blood counts (CBCs) are a routinely used laboratory test, and the distribution of leukocyte types such as the neutrophil to lymphocyte ratio (NLR) as a prognostic inflammatory marker in vascular diseases or the lymphocyte to monocyte ratio (LMR) as a marker of outcome in malignancies has gained attention [1,2,3,4].

In addition, several scores have been used to evaluate cancer patients at high risk for thromboembolic events, although inflammation may play an additional role in this direction. Nevertheless, bleeding risk should also guide clinical decision-making and is of high importance in patients before starting oncological treatment. A few clinical studies have evaluated the association of non-specific inflammatory markers—NLR, platelet to lymphocyte ratio (PLR), or lymphocyte to monocyte ratio (LMR)—and risk of bleeding [5,6].

The IMPROVE score is a validated tool for assessing patients for the three-month risk of thrombosis, although the IMPROVE Bleeding score is lacking researchers’ attention [7,8]. The prognostic role of NLR in tumor progression and disease recurrence has been highlighted in previous studies, but to the best of our knowledge in cancer patients, few have investigated the association of non-specific pro-inflammatory markers derived from CBC and bleeding risk [9,10]. Therefore, we aimed to evaluate the relationship of NLR, LMR, and the risk of thrombosis by using the IMPROVE score as well as the risk of bleeding using the IMPROVE Bleeding Risk Assessment Score in patients with non-muscle-invasive bladder tumors.

## 2. Material and Methods

We performed a retrospective observational study at the County Clinical Hospital, Department of Urology in Targu Mures, analyzing NMIBC patients between January 2016 and December 2020. The study was conducted in accordance with the World Medical Association Declaration of Helsinki and was approved by the Ethical Committee of the Mures County Hospital (Opinion form Nr 9572). A total of 130 patients with non-muscle-invasive bladder cancer were included in this survey who met the inclusion criteria: age > 18 years, stage pTa-pT1 NMIBC. Exclusion criteria were the following: age < 18 years, stage pT2 or higher, and presentation of metastasis, inflammatory, liver or autoimmune diseases, or other systemic neoplasms. In order to evaluate the risk of thromboembolic events as well as those of bleeding, the IMPROVE and IMPROVE Bleeding Risk Assessment Scores were calculated for each patient by using the calculators available online. Tumor staging was established before inclusion in the study. We noted the grade, stage and size of the tumor. Based on the cut-off value of 4 points for the IMPROVE score (modified IMPROVE—Spyropoulos), subjects scoring < 4 points were categorized into the Low IMPROVE group, whereas those who scored ≥ 4 points were in the High IMPROVE group. 

Anthropometric, clinical, and biological data were recorded for each patient. Blood samples were collected in the morning between 8 and 10 AM from the brachial vein after at least 8 h of fasting. As part of a routine laboratory test, a complete blood count was analyzed at a maximum of 7 days prior to surgical intervention. To assess the inflammatory status of the patients, neutrophils, lymphocytes, monocytes, neutrophil to lymphocyte ratios, and lymphocyte to monocyte ratios were calculated. NLR > 3 was considered as a negative prognostic marker in NMIBC; therefore, we also evaluated IMPROVE and IMPROVE Bleeding scores among subjects with low NLR values < 3 and high NLR values ≥ 3.

Statistical Analyses

The statistical analysis was performed using the IBM SPSS Statistics 22 program (IBM Corporation, Armonk, NY, USA), involving descriptive and interferential statistics. Quantitative variables were tested by using the Kolmogorov–Smirnov test, and are expressed as median values, and between-group comparisons were performed using the Student’s *t*-test or the Mann–Whitney U test. Categorical variables were displayed as frequencies, *n* (%), and comparisons between the groups were assessed with the chi-squared test. We also analyzed the correlation between the variables by using Spearman’s test. Variables included in the multivariate analysis were selected on the basis of the best results of bivariate analyses (at a significance level of *p* < 0.05). Univariate and multivariate regression of predictive factors was conducted using the IMPROVE score as a dependent variable and age, lymphocytes, NLR (>3), monocytes, lymphocyte to monocyte ratio, tumor size > 3 cm and previous VTE as independent predictive variables. Statistical significance was set at *p* < 0.05.

## 3. Results

Among 130 (100%) subjects included in the study, the Low IMPROVE group included 66 patients, whereas in the High IMPROVE group there were 64 patients. Characteristics of the groups, as well as laboratory data and tumor grading, are presented in Table 1. 

The distribution of the subjects based on gender was as follows: Twenty-one females and 109 males. Regarding the tumor stage, in females, 14 had stage pTa, whereas 7 had stage pT1. In males, 71 had stage pTa and 38 had stage pT1; this was not statistically significant, *p* = 0.89. We noticed a statistically significant difference among males and females regarding the IMPROVE score, 3.95 ± 0.11 versus 2.9 ± 0.27, *p* = 0.006. The IMPROVE Bleeding score was 9.69 ± 0.21 in males versus 8.69 ± 0.61 in females, *p* = 0.07. 

In a univariate logistic regression model (Table 2), the IMPROVE score was significantly influenced by the lymphocyte count ((OR: 0.12; CI95%: 0.05–0.26, *p* < 0.0001), NLR ratio (OR: 0.37; CI95%: 0.18–0.76, *p* = 0.007), monocyte count (OR: 8.91; CI95%: 1.99–39.76, *p* = 0.0042) and lymphocyte to monocyte ratio (OR: 0.40; CI95%: 0.29–0.56, *p* = 0.0001).

In the multivariate logistic regression model (Table 3), IMPROVE score is significantly influenced by the lymphocyte count and the neutrophil to lymphocyte ratio. 

Between the variables of interest we applied Spearman type correlations, obtaining significant values, *p* value below 0.05 (*. Correlation is significant at the 0.05 level (2-tailed)). Respectively below 0.01 (**. Correlation is significant at the 0.01 level (2-tailed)). The correlation coefficient r (rho) is interpreted compared to the value 1, at values below 1 we have negative correlations, at values above 1 there are positive correlations. The data are represented in Table 4.

## 4. Discussions

In this study, we investigated the relationship between CBC-derived parameters such as NLR, LMR and the risk of thrombosis, as well as those of bleeding in patients with non-muscle-invasive bladder cancer hospitalized for surgical intervention. The incidence of venous thromboembolic events is much higher in patients with bladder cancer compared to other malignancies; thus, we considered it of high importance to investigate this population [11]. The non-muscle-invasive type represents more than 70% of diagnosed bladder cancers; therefore, it was well-founded to investigate thromboembolic and bleeding risks in this cohort [12].

Although the IMPROVE score is used in patients in order to evaluate the three-month risk of venous thromboembolic events, patients with bladder cancer admitted to the hospital for various medical or non-medical conditions often present several coagulation-related problems which have to be solved by a multidisciplinary team [13,14,15].

Our results revealed that patients with higher IMPROVE risk scores are associated with older age, higher NLR and monocyte values, as well as lower LMR values. Among these subjects, a history of deep vein thrombosis was recorded in 28% compared to 5% in the Low IMPROVE group, and anticoagulation therapy during admission was required in 78.12%. However, subjects with NLR values greater than 3 are at high risk for the progression and recurrence of cancer; therefore, we thought it would be meaningful to assess both the thromboembolic as well as the bleeding risk. We noticed greater values for IMPROVE and IMPROVE Bleeding scores; however, we could not demonstrate statistically significant correlations between clotting disorders and NLR values. These results are in accordance with some data available in this research field [1]. Despite these results, the NLR has been already validated as a prognostic marker in the recurrence and progression of bladder tumors, although there is a need for further research to identify other CBC-derived parameters as risk factors for thromboembolic events [16].

Another study performed on subjects admitted for pulmonary embolism revealed that increased levels of NLR and LMR were associated with increased mortality risk in patients with moderately low and low risk of PE [17]. In cardiovascular as well as in oncological diseases, some non-specific inflammatory biomarkers—NLR and LMR—have gained attention due to their power in predicting the occurrence of cancer-related deep vein thrombosis [1,18,19,20]. Although the cut-off value for LMR differs in each study, its role as a prognostic factor in cancer, including bladder cancer, has been well established [21,22,23,24]. Thus far, studies have revealed that the imbalance between lymphocytes and monocytes worsens the inflammatory status of the patient, which will lead to thrombosis. However, there is no clear cut-off value defined for LMR; in a retrospective study, Zhu et al. found that LMR has the power to predict the risk of deep vein thrombosis in patients after total joint arthroplasty. The cut-off value was based on an assessment of the area under the curve [5]. Li et al. found that low LMR was linked to poor outcomes in patients with cerebral venous sinus thrombosis, which could be the result of monocyte deactivation [25].

In our study, in uni- and multivariate regression models, we demonstrated that the IMPROVE score is significantly influenced by the lymphocyte count as well as high NLR values. Since, lymphocyte count does not show Gaussian distribution, it has been displayed as minimum to maximum, but the same parameter assessed as mean ± SD the difference is statistically significant, 2.04 ± 0.89 versus 1.65 ± 0.83 × 10^3^/µL (*p* < 0.005) in the high IMPROVE group. To the actual state of knowledge, lymphocyte count alone does not have the power to be an independent prognostic factor for venous thromboembolism; however, as part of the neutrophil to lymphocyte ratio is has been proved to be an inflammatory marker of tumor progression and thus of thrombosis.

In a comprehensive study, Braun A. et.al. stated that platelets play a critical role in thrombosis, hemostasis and the arrest of bleeding in healthy conditions [26], and thrombotic events have frequently been observed in cancer patients, indicating an active involvement of platelets and factors released from platelets in tumor progression, enhancing pro-coagulant activity and blood clotting [27,28]. Although the systemic effects of platelets in thrombotic complications of cancer patients have been described, compelling experimental and clinical evidence have linked platelet function to tumor angiogenesis, tumor progression and metastasis through the interaction of platelets with cancer cells and the tumor microenvironment [26].

Ho-Tin-Hoé B. et.al. published an article presenting that platelets support the recruitment of leukocytes to sites of inflammation and that platelets not only serve in concert as building blocks of the hemostatic plug, but also act individually as gatekeepers of the vascular wall to help preserve vascular integrity while coordinating the host defense, introducing the notion of “inflammation-associated hemostasis” [29].

The presence of cancer and lack of physical activity for at least three days as promoting factors for deep vein thrombosis can be found in several scores assessing the risk of deep vein thrombosis in hospitalized patients [30,31]. In cancer patients, inflammatory markers have been implemented in the guiding of specific therapy; thus, the same markers could be of high importance to also assess the risk of thrombosis. Therefore, the role of LMR as a new thrombo-inflammatory marker has been established in earlier studies, and seems to predict future thrombosis [25]. These findings were also supported by our study, where patients with a high risk of thrombosis, defined by an IMPROVE score greater than 4, were associated with lower values of LMR. By analyzing the characteristics of the study group based on gender, we noticed that only 16.3% were females. Nevertheless, the incidence of tumor stage was almost the same: 66.66% pTa in females versus 65.13% in males. The small number of females participating in this survey could also explain the lower scores of IMPROVE and the IMPROVE Bleeding scores. However, a statistically significant difference was recorded only in the case of the IMPROVE score.

The COVID-19 pandemic has represented an unprecedented challenge to every health system, putting an extra burden on the management of neoplasms. Matteo F. et al., in a multi-institutional retrospective cohort analysis on 2591 patients, stated that delays in treatment schedules and disease management were observed, and investigation of the oncological impacts of those differences should be advocated [32].

The study has several limitations. Firstly, the lack of data regarding D-dimers could have increased the statistical significance of the IMPROVE score and other proinflammatory markers. Other prothrombotic factors such as body mass index or tobacco use are also lacking in data, although we aimed to only focus on CBC-derived markers with a potential role in predicting venous thromboembolic events. Secondly, we did not include a control group consisting of patients with acute thrombosis or healthy individuals without cancer. The distribution of genders was not equal, because in our database, the male gender was more represented. We found that patients in the High IMPROVE group associated lower lymphocyte count which was also stipulated to influence the IMPROVE score in the univariate logistic regression model. However, we cannot state yet that lymphocyte count alone is a prothrombotic marker, only if integrated in the formula of other CBC-derived parameters. We think that future researches should implement other markers of venous thromboembolism like D-dimers or specific tumor markers in order to increase statistical significance of these data.

## 5. Conclusions

In our study, we found that subjects with NMIBC who associated low lymphocyte count, an NLR greater than 3, and a low lymphocyte to monocyte ratio were at higher risk of developing venous thromboembolic events, as reflected by an IMPROVE score greater than 4. The CBC-derived lymphocyte count, and neutrophil to lymphocyte ratio, are low-cost prothrombotic markers, useful in daily clinical practice in order to identify patients with cancer at high risk of developing venous thromboembolism in the perioperative period.

The IMPROVE and the IMPROVE Bleeding Risk Assessment Scores are easy to use, and if completed by the CBC-derived neutrophil to lymphocyte ratio and lymphocyte to monocyte ratio, could be of practical relevance in the decision of starting prophylactic anticoagulation therapy during hospital admission.

## Figures and Tables

**Table 1 jcm-10-05270-t001:** Demographic and laboratory data of the study groups.

Characteristics	Low IMPROVE Group <4 Points (*n* = 66)	High IMPROVE Group ≥4 Points (*n* = 64)	*p*-Value
Age (years) mean ± SD	65.77 ± 11.316	73.64 ± 9.03	<0.0001
Gender, M/F nr (%)	55/11 (16.9%)	54/10 (15.6%)	0.84
Lymphocytes, 10^3^/µL median (min-max) and mean± SD	2.00 (0.00–4.00) 2.04 ± 0.89	2.00 (0.00–4.00) 1.65 ± 0.83	0.005
Neutrophils, 10^3^/µL median (min-max)	6.14 ± 2.88	7.22 ± 4.6	0.21
Neutrophil to lymphocyte ratio median (min-max)	2.44 (1.15–19.68)	3.50 (1.05–63.47)	0.002
NLR > 3, nr (%)	26 (40.0%)	41 (64.1%)	0.008
Monocytes, 10^3^/µL median (min-max)	1.00 (0.00–1.00)	0.00–2.00	0.046
Lymphocyte to monocyte ratio median (min-max)	3.75 (0.557–12.42)	2.69 (0.32–8.31)	0.0001
Previous VTE, nr (%)	2 (5.0%)	18 (28.1%)	0.004
Tumor stage pT1, nr (%)	20 (30.8%)	24 (37.5%)	0.28
pT1 + CIS	0 (0.0%)	1 (1.6%)	
pT1a	0 (0.0%)	1 (1.6%)	
pTa	43 (66.2%)	38 (59.4%)	
pTa + CIS	2 (3.1%)	0 (0.0%)	
Tumor grade G1	3 (4.6%)	1 (1.6%)	0.43
G2	34 (52.3%)	31 (48.4%)	
G3	23 (35.4%)	22 (34.4%)	
Tumor size > 3 cm, nr (%)	44 (67.7%)	54 (84.4%)	0.047

VTE—venous thromboembolism, CIS—cystectomy.

**Table 2 jcm-10-05270-t002:** Results from univariate logistic regression.

Variables	Odds Ratio	95% CI	*p*
Age	1.0200	0.9873 to 1.0538	0.2329
Lymphocytes	0.1204	0.0546 to 0.2653	<0.0001
NLR (>3)	0.3740	0.1834 to 0.7624	0.007
Monocytes	8.9106	1.9967 to 39.7642	0.0042
Lymphocyte to monocyte Rratio	0.4082	0.2925 to 0.5696	0.0001
Tumor size >3 cm	1.3258	0.5480 to 3.2073	0.53
Previous VTE	0.9091	0.3427 to 2.4114	0.84

**Table 3 jcm-10-05270-t003:** The final multivariable logistic model.

Variables	Odds Ratio	95%CI	*p*
Lymphocytes	0.1214	0.0303 to 0.4864	0.0029
NLR (>3)	0.4697	0.2170 to 0.9845	0.0098
Monocytes	12.8667	0.2403 to 688.8504	0.2084
Lymphocyte to monocyte ratio	0.8590	0.4312 to 1.7111	0.6655

**Table 4 jcm-10-05270-t004:** Spearmen’s correlation for study population with IMPROVE score ≥4 points.

Correlations								
			Age	Neutrophils	Lymphocytes	NLR	Monocytes	LMR
Spearman’s rho	Age	Correlation Coefficient	1.000	−0.070	−0.196	0.110	−0.009	−0.269 *
		Sig. (2-tailed)	.	0.575	0.111	0.376	0.944	0.028
	Neutrophils	Correlation Coefficient	−0.070	1.000	0.138	0.678 **	0.159	−0.254 *
		Sig. (2-tailed)	0.575	.	0.266	0.000	0.198	0.038
	Lymphocytes	Correlation Coefficient	−0.196	0.138	1.000	−0.489 **	0.307 *	0.466 **
		Sig. (2-tailed)	0.111	0.266	.	0.000	0.012	0.000
	NLR	Correlation Coefficient	0.110	0.678 **	−0.489 **	1.000	−0.023	−0.613 **
		Sig. (2-tailed)	0.376	0.000	0.000	.	0.851	0.000
	Monocytes	Correlation Coefficient	−0.009	0.159	0.307 *	−0.023	1.000	−0.408 **
		Sig. (2-tailed)	0.944	0.198	0.012	0.851	.	0.001
	LMR	Correlation Coefficient	−0.269 *	−0.254 *	0.466 **	−0.613 **	−0.408 **	1.000
		Sig. (2-tailed)	0.028	0.038	0.000	0.000	0.001	.

* Correlation is significant at the 0.05 level (2-tailed). ** Correlation is significant at the 0.01 level (2-tailed).

## Data Availability

Data sharing not applicable.

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
