# Peer review of "Inflammatory Markers and Thromboembolic Risk in Patients with Non-Muscle-Invasive Bladder Cancer"

_jcm, 2021, doi:10.3390/jcm10225270_

Round 1

Reviewer 1 Report

The study of Balan D. et al. is very interesting. The aim is to implement the evaluation of thrombotic and haemorrhagic risk with parameters deriving from the patients' a complete blood count. In patients with bladder cancer this detailed assessment become necessary as patients with bladder cancer are both at risk of thrombosis and at risk of bleeding due to the tendency of bladder lesions to bleed generating micro and gross haematuria, both spontaneously and after TURBT. In this population, the evaluation of the balance of thrombosis and haemorrhage is certainly an important clinical aspect. The study, with important limitations in terms of number of enrolled patients and type (retrospective), is interesting as a preliminary evaluation of the haematological parameters in this patient cohort.

The manuscript is interesting but it needs of major and minor revisions:

Major revision:

  • IMPROVE is a tool validated for hospitalized medical patients. You used it in patient cohort that is not medical nor hospitalized. Is there a study that validated this tool in patients with malignancies? If not, can you explain the reasons that led you to choose this tool and exclude other?
  • You used numerous parameters derived from complete blood count. When you collected the data? When they were hospitalized before or post TURBT or in outpatient setting? They had gross haematuria? These factors are important to define because in post-TURBT or in case of gross haematuria the values of lymphocyte, neutrophils and monocytes may be different from outpatient setting without gross haematuria.
  • While I was reading the manuscript, I had the feeling that the contents were scattered and in disorder. I also had the feeling that a lot of data is missing. In “Results” can you use paragraphs? For example “Low and High IMPROVE group – NRL”, “Low and High IMPROVE group – LMR”, “IMPROVE Bleeding risk – NRL” and “IMPROVE Bleeding risk – LMR”. Currently in your paper are missing data on: NRL correlation with IMPROVE score, LMR correlation with IMPROVE score and NRL/LMR correlation with IMPROVE Bleeding risk. I think that you need revised the section “Results”. Moreover in “Conclusions” you state that male gender associated “with higher risk of thrombosis and those of Bleeding (186-187)” but in line 116 (Results) you stated that “. The bleeding risk score was 9.69 ±0.21 in males versus 8.69±0.61, p=0.07”, without significant difference.
  • Revise please English language, primarily in “Conclusions”.

Minor revisions:

  • You should report the BMI in the anthropometric, clinical, and biological data. It was reported in many studies as a risk factor of venous thromboembolism and it could be a confounding factor in your analysis.
  • You should clarify on text which type of thrombosis are you analysing. “It is well known, that patients with malignancies have a high risk of thrombosis (line 40).”  Arterial or venous thrombosis?
  • In Figure 1, is Spearman, not Spearmen and the figure is not cited and commented in the text.
  • In the abstract there is not the type of study (retrospective observational study)
  • In Table 1, you should delete “years” from line 2/columna 3. About the “lymphocytes, absolute ratio” (line 5) and “Neutrophils, absolute ratio” (line 6), what did you mean with absolute ratio? It seems to me that they refer to mean value ±SD of lymphocytes and neutrophils and you need specify the unit of measure. Same concept for “Monocytes” (line 7). Moreover, you write in Statistical Analyses that “Categorical variables were displayed as frequencies, n (%)” but in Table 1 for “Previous DVT” and “Tumour recurrence” you show only % and you should report also the absolute number.

Reviewer 2 Report

In their study, Balan et al., studied bleeding risk assessment scores, and compared the neutrophil-lymphocyte ratio, lymphocyte monocyte ratio and risk of thrombosis in patients with non-muscle invasive bladder cancer. They found that in cancer patients with high NLR values were associated with a higher risk of thrombosis as well as those of bleeding. Although the study is interesting, however, the manuscript needs to be significantly improved. The authors should discuss the difference between normal hemostasis and inflammatory hemostasis. Platelets are major players of hemostasis and thrombosis. The authors could discuss the role of platelets in bladder cancer, platelet markers, known as save antithrombotic targets, how to deal with the balance between hemostasis and thrombosis, some targets are known to have inhibitory effects on thrombosis while preserving normal hemostasis. Following articles can be useful. Braun et al., Front Oncol 2021 Jul 12;11:665534. doi: 10.3389/fonc.2021.665534, Ho-Tin-Noé, Blood 2018 Jan 18;131(3):277-288. doi: 10.1182/blood-2017-06-742676. Platelets are involved in inflammatory bleeding as well. When neutrophils migrate to inflammatory sites the cross vessels, this process induces bleeding, and also thrombosis, a process called thrombo-inflammation. These processes can be involved in bladder cancer as well, regarding the results that the authors found. I suggest that the authors improve their manuscript by providing a synthetic and critical discussion about possible molecular mechanisms.

Round 2

Reviewer 2 Report

The authors addressed my comments. 

Author Response

-